# Ecological niche measurement and high-quality development of "the Belt and Road" core area

**Hang Zhang**, **Nurguli Abdusuli** *

School of Economics and Management, Xinjiang Agricultural University, Urumqi, China

* 453244794@qq.com

**Data Availability Statement:** All relevant data for this study are within the paper and its Supporting Information files. Raw third party data is available upon request from Statistics Bureau of Xinjiang Uygur Autonomous Region website (https://tjj.

## Abstract

A new stage in promoting the construction of the Silk Road Economic Belt Core Area, and Xinjiang has been transformed from a relatively closed inland area into an open border. In order to promote the high-quality development of Southern Xinjiang and solve the imbalance contradiction between the development of the Northern Xinjiang and Southern Xinjiang, taking the four districts in Southern Xinjiang as the study area, constructing a high-quality development ecological niche index system of three levels, namely economic, social and ecological, adopting the entropy method to assign weights to the evaluation indexes, and measuring the ecological niche width and the degree of ecological niche overlap of this region in the period from 2011 to 2020. The results show that: Firstly, tourism has the greatest impact on the ecological niche of economic development in state N, with a weighting of 14.18%; Secondly, the ecological status width of economic development in state N demonstrates a structural characteristic of "low level and low gap". The average value of ecological niche width is at class III, indicating a low development status and weak regional influence; Thirdly, the ecological niche overlap of state N is significantly influenced by spatial factors. Regions Z and S are closer together, resulting in higher competition for resource utilization and an average ecological niche overlap at class II. The other two regions are at class III. According to the theory of ecological niche expansion and separation, a specialization separation strategy should be adopted for areas with "low width and high overlap", and a strengthening expansion strategy should be adopted for areas with "low width and low overlap", to optimize the structure of ecological niches and promote high-quality development of the region.

## 1. Introduction

Since the proposal of the Silk Road Economic Belt 10 years ago, Xinjiang has demonstrated its irreplaceable status and role in its construction through practical contributions. On 26 August 2023, Xi Jinping emphasized the importance of fully utilizing Xinjiang's unique location advantages, actively integrating into the new development pattern, and accelerating the construction of the core area of "the Belt and Road" from a practical perspective after listening to a

xinjiang.gov.cn/). The authors confirm they did not have any special access privileges that others would not have when attempting to access the minimal data used from this third party source.

**Funding:** This paper is funded by the National Social Science Fund of China: Study on the Cultural Coexistence Paradigm of Embeddedness Multi-ethnicof Villages in the Context of "Culture Moistening Xinjiang", grant number 22BMZ123, which provided financial support for our data collection and investigate activities.

**Competing interests:** The authors have declared that no competing interests exist.

report on Xinjiang's work. With the promotion of the Belt and Road initiative, Xinjiang has transitioned from a relatively closed inland region to an open frontier [1]. However, the high-quality development of Xinjiang has been hindered by the problem of development imbalance and incoherence between the north and south borders [2]. While acknowledging the disparity between the overall economic development of Southern Xinjiang and its regional volume, it is important to note that the region is abundant in energy resources, boasts unique location advantages, and has significant potential for special industries [3]. Additionally, it has clear advantages in providing aid to Xinjiang and possesses the necessary foundation, conditions, potential, and advantages to achieve high-quality development. Therefore, based on the current development trends, the Southern Xinjiang region needs to clarify its development status, assess resource utilization, and provide theoretical support to narrow the gap between Northern and Southern Xinjiang and achieve high-quality development.

Regional economic development has been assessed as a multi-level comprehensive assessment in existing studies. Early economic research in China was mostly used to study the speed of economic development. With the guidance of the green development concept, ecological civilization construction and other policies, China is now in a transition period from high-speed development to high-quality development [4–7]. High-speed economic growth does not fully represent high-quality economic development, and short-term high-speed growth may cause social problems, such as resource or environmental problems [8], and the intensification of polarisation between the rich and the poor [9]. It therefore focuses on the greening of the economy [10–15] or sustainable development [16–20] now. These are included under the heading of high-quality development. The prevailing view is that the development is a complex system. Regional economic development is influenced by a variety of factors and is a comprehensive expression of the economy, society and ecology of a region [21–23]. No single indicator can fully reflect the level of economic development, it should be scaled with multi-level indicators that meet the needs of the assessment. Accordingly, most of the current studies are comprehensive assessments of the level of regional economic development, which are not limited to economic indicators, but also include indicators measuring demographic, social, ecological and environmental aspects [15, 24–30]. Based on the continuous deepening of research on regional economic development, scholars believe that economic development is a continuous evolutionary process from low to high level and from quantitative change to qualitative change [31]. During the evaluation process, it is essential to consider both "state" and "trend" dimensions for accurate analysis. It is aligning with the theory of ecological niche, which comprehensively examines the evaluated object through the integration of its states and trends [32]. The niche ecostate-ecorole theory fit these dual dimensions well, so some scholars apply ecological niche theory in the study of regional economy to the evaluation and analysis of the level of regional economic development, the competitiveness of regional economy and regional competitions and partnerships, etc. These studies have mainly focused on coastal or inland areas, while there are very few ecological niche assessments of economic development for border areas.

The concept of the ecological niche comes from ecology, first defined by Joseph Grinell [33] in his study of organisms in nature, and since then foreign scientists have continued to add to its meaning [34, 35]. After almost a century of evolution, the notion of the "ecological niche" now exceeds the scope of ecological studies and is being progressively extended to other domains including economic [36–38], tourism [39–41], cultural [42–44], technological [45, 46], and industrial [47] niches. The application of ecological niche theory in the field of regional economics is relatively recent, but the model of analogy between regional units and species units has been widely used in the study of regional geography and regional economics [48–53]. The ecological niche theory is applied to the study of economic development evaluation and the ecological niche concept of economic development is defined as: the spatial and

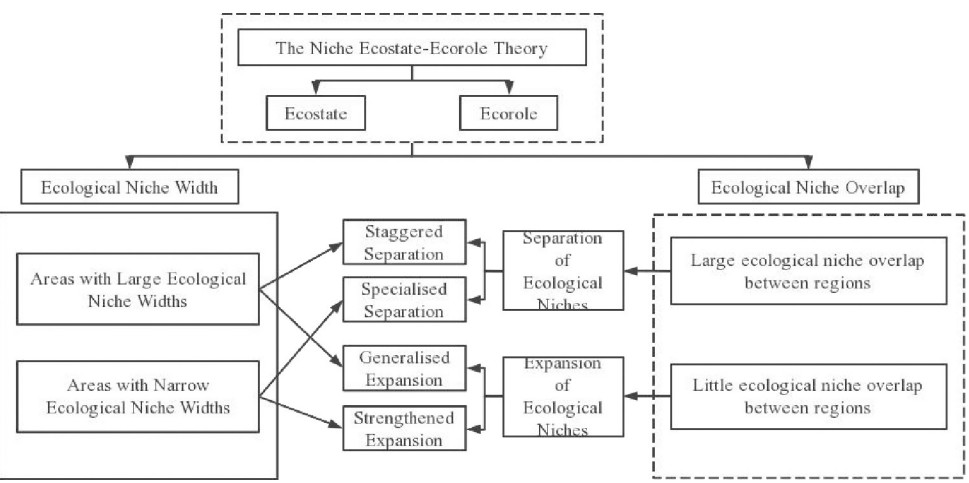

**Fig 1. Ecological niche theory relationship diagram.**

temporal space occupied by different ecological units in the ecological system of regional economic development under the influence of many factors such as social, economic and environmental factors, as well as the relative status and role of that ecological unit in relation to other ecological units. The ecological niche theory encompasses primarily the ecological niche ecostate-ecorole, the ecological niche width, the ecological niche overlap, the expansion and separation of ecological niche, the relationship of these theories is shown in Fig 1.

Regional economic development is a cohesive entity that arises from the interplay between internal and external factors associated with economic growth in a specific locality, either directly or indirectly [54]. From a multi-dimensional spatial and temporal perspective, the regional economic development system is a complex ecosystem with multiple relationships, dimensions and levels. The development process itself moves from low to high levels and from quantitative change to qualitative change [6, 7]. Therefore, when assessing the progress of regional economic development, it is crucial to consider both the "state" and "process" aspects, in line with the niche ecostate-ecorole theory. Using the theory of ecological niche to assess economic development, the term "ecostate" describes the present state of regional economic development, while "ecorole" refers to the rate of renewal and growth, indicating a trend towards change.

Ecological niche width and degree of overlap are quantitative expressions of the position or degree of competition of each unit in the regional economy, based on the concepts of "ecostate" and "ecorole" of ecological niches. On that basis, the calculation of ecological niche width is applied to the study of economic development, and the larger the ecological niche width of an ecological unit, the higher the position of the ecological unit in the ecosystem of economic development [55]. Ecological niche overlap is the root cause of competition between ecological units [56], and refers to the phenomenon in an ecosystem where two ecological units with the same or similar ecological niche compete for the same resources in the same space [57]. A quantitative study of the degree of competition for economic development between regions, reflecting the competition for resources within the regional economic development system by calculating ecological niche overlap between different regions.

Ecological niche expansion or separation is a strategy for regions to adjust their ecological niche width and overlap characteristics, promoting sustainable economic development through various means. Ecological niche expansion in economic development is the process of increasing regional competitiveness and dominance to improve or maintain economic development within the current conditions of the regional economic system, achieved by expanding the

region's ecological niche. A minor overlap within a region is more probable to lead to ecological niche expansion. This is due to inefficient usage of economic development resources, requiring ecological niche expansion. This is manifested in two ways. First, generalized expansion. The higher ecological niche width for economic development means that the economic development of the region is more mature, and the resources available for development in the region are limited, so the ecological units in the system evolve towards "omnivory" or "generalization" in order to break the bottleneck of economic development [58]. Second, strengthened expansion. Since economic development is relatively weak in areas with small ecological niche width, and the small width of economic development limits the availability of various resources within the region, it is necessary to increase the level of regional economic development by exploring potential ecological niches or introducing new ecological niches. Ecological niche separation refers to differences in the selection of ecological niches across different regions to decrease competition for resources in numerous dimensions [59]. Ecological niche separation arises in regions where there is substantial overlap, and considerable overlap in economic niche indicates similarities in resource use or development across various parts of the area. This competition for resources and regions' exclusion suggest a requisite for ecological niche separation. There are two specific scenarios. First, specialized separation. Areas with narrower ecological niches have a less favourable foundation for economic advancement and hold a lower position within the regional economic development scheme. To prevent the undue expansion of ecological niches in other regions placing significant pressure on its own economic development, it is imperative to concentrate on the growth of its own favourable projects to sidestep direct competition with other regions. Second, staggered separation. Regions with broad ecological niche widths possess a strong potential for economic growth and ample resources accessible through the regional economic development system. Although there is competition present, ecological niches can be dislocated by distinguishing spatial, temporal or resource-use niches, resulting in a complementary economic development approach rather than specialization.

Based on the characteristics outlined above, ecological niche theory can combine both the current state and trend to analyse the level of economic development. Additionally, its rigorous theoretical framework can propose appropriate ecological niche strategies for the characteristics of different regions. Currently, the core area of the Belt and Road is experiencing rapid development, and the ecological niche perspective is well-suited for assessing local economic development. Therefore, this paper takes the state N in South Xinjiang as a case study and applies the ecological niche theory to evaluate the economic development of this area. The state N is regarded as an economic ecosystem, and the regions K, Z, S and T are regarded as ecological units in the system, and the ecological niche widths and overlapping degrees of the four regions are calculated by the ecological niche model, so as to evaluate the status of each region and the degree of competition in the state N, so as to provide theoretical references and data support for the economic development of the state N.

## 2. Materials and methods

### 2.1. Ecological niche model

**2.1.1. The ecological niche width model.** The formula for calculating the combined width of the ecological niche of economic development is:

$$M_i = \frac{\sum\limits_{\alpha=1}^{m} N_{i\alpha} w_{\alpha}}{\sum\limits_{\alpha=1}^{m} w_{\alpha}}, \tag{1}$$

in Eq(1): $\alpha = 1,2,\ldots,m$ denotes the number of indicators, $M_i$ denotes the ecological niche width of economic development of the ecological unit, and $w_\alpha$ denotes the weight of each indicator variable, the magnitude of which reflects the degree of influence of each indicator factor on the ecological niche of economic development, and can reflect the degree of importance of each indicator factor. The ecological niche width for a single metric ($N_{i\alpha}$) is calculated using the formula:

$$N_{i\alpha} = \frac{\frac{(S_{i\alpha} + AR_{i\alpha})}{n}}{\sum_{i=1}^{n}(S_{i\alpha} + AR_{i\alpha})}, \tag{2}$$

in Eq (2): $i = 1,2,\ldots,n$ denotes the number of ecological units, and $N_{i\alpha}$ denotes the width of the ecological niche of the ecological unit $i$ on the indicator $\alpha$; $S_{i\alpha}$ and $R_{i\alpha}$ represent the ecostate value and ecorole value of the ecological unit on the indicator, respectively, where the ecostate value is the data after the current data of each indicator is dimensionless, and the ecorole value is the data after the growth rate of each indicator is dimensionless; A is the scale conversion factor, which is taken as 1 because the time span for calculating the growth rate is 1 year. The interval of the ecological niche width is [0, 1], and the closer it is to 1, the higher the economic development status of the ecological unit and the greater its regional economic influence or dominance, and vice versa.

**2.1.2. The ecological niche overlap model.** The formula for calculating the degree of overlap of ecological niches between two regions is as follows:

$$b_{ij} = \frac{\sum_{\alpha=1}^{m} N_{i\alpha} N_{j\alpha}}{\sqrt{\sum_{\alpha=1}^{m} N_{i\alpha}^2 \sum_{\beta=1}^{m} N_{j\alpha}^2}}, \tag{3}$$

in Eq (3): the ecological niche overlap between areas i and j can be denoted by $b_{ij}$; the ecological niche widths of areas i and j on indicator $\alpha$ can be represented by $N_{i\alpha}$ and $N_{j\alpha}$, respectively, whilst d indicates the straight-line distance between the two areas' spatial centres. The range for the ecological niche overlap is [0, 1], with 0 indicating complete ecological niche separation and 1 indicating complete ecological niche congruence.

In addition, based on the principle of distance decay, this paper posits that the level of overlap between regions is influenced by the distance separating the two entities involved. The greater the distance between two regions, the less likely they are to compete for the same resource. Therefore, spatial factors need to be taken into account when calculating the ecological niches overlap for economic development. The gravitational model is introduced in order to adjust the extent of overlap among ecological niches such that it aligns more accurately with practical economic principles.

$$B_{ij} = G\frac{b_{ij}}{d_{ij}}, \tag{4}$$

in Eq (4): $B_{ij}$ represents the improved ecological niche overlap, while $d_{ij}$ indicates the linear distance between the area i and j. G is a fixed parameter set to 1.

This paper assumes that third parties do not affect the degree of overlap between the two regions. Therefore, a mean method is utilized to compute the collective overlap of the

ecological niches for economic progress in the area:

$$\delta_i = \frac{\sum_{j=2}^{n} B_{ij}}{n-1},$$ (5)

in Eq (5): $\delta_i$ represents the degree of overlap of the region's integrated ecological niche for economic development.

## 2.2. Establish an evaluation index system for ecological niche of economic development

**2.2.1. Selection of indicators.** Regional economic development aims to achieve coordinated growth across the economic, social, and ecological dimensions, in order to fulfil the public's desire for an improved quality of life. In the economic dimension, high-quality development should be dynamic and sustainable, not only in terms of steady growth in economic output, but also in terms of sustainable optimization of the economic structure and the emergence of new industries. In the social dimension, high-quality development ought to facilitate the populace's enjoyment of development benefits, foster social justice on the premise of people's living standards' continual betterment, and reduce the urban-rural disparities. In the ecological dimension, the concept of green development and the synergistic development of economy and ecology should be fully reflected in order to achieve high-quality development, which includes reducing pollution, lowering energy consumption and creating an ecological environment worth living in. Therefore, this paper presents a summary of the indicator system used to evaluate China's recent economic development. It selects commonly-represented indicators and measures the overall strength of regional economic development in state N from three perspectives: economic, social, and ecological. Subsequently, it identifies 26 indicators, taking into account data accessibility, and details them in Table 1.

**2.2.2. Allocation of indicator weights.** The entropy value method provides an effective means of dealing with multiple indicators, uncertainty information, and broad applicability, making it an accessible and practical approach. Thus, this paper applies the entropy value method to weight indicators of ecological niche of economic development. The raw data for the indicators were first standardized by the extreme value method:

$$r = \frac{x - x_{min}}{x_{max} - x_{min}},$$ (6)

in Eq (6): r represents the standardized processed value, whilst x represents the original data, $x_{min}$ denotes the minimum value within the dataset, and $x_{max}$ denotes the maximum value within the dataset. Next, calculate the entropy of the indicator:

$$e_\alpha = \frac{1}{\ln n} \sum_{i=1}^{n} p_i \ln p_i,$$ (7)

in Eq (7): $e_\alpha$ denotes the proportion of data processed using entropy within this data set, whilst $p_i$ represents the entropy of the initial indicator. Finalize the weights

$$w_\alpha = \frac{1 - e_\alpha}{\sum_{\alpha=1}^{m}(1 - e_\alpha)},$$ (8)

in Eq (8): the final weight of each indicator variable is represented by $w_\alpha$.

**Table 1. Ecological niche evaluation system for economic development.**

| System Layer | Normative Layer | Indicator Layer | Nature of Indicators | Description of Indicators |
|---|---|---|---|---|
| Economic Subsystem | economic aggregate | total regional production | + | Measuring the total level of the economy. |
| | | fixed-asset investment | + | Measuring the level of social investment. |
| | | unit yield of food crops | + | Measuring the level of agricultural development. |
| | | industrial output per capita | + | Measuring the level of industrial development. |
| | | tourism revenue | + | Measuring the level of tourism development. |
| | economic structure | share of tertiary output in total production | + | Measuring industrial structure. |
| | | total retail sales of consumer goods as a share of total production | + | Measuring the distribution structure. |
| | | percentage of population in agriculture | - | Measuring labour force structure. |
| Social Subsystem | living standards of the population | disposable income of the population | + | Measuring the level of income of the population. |
| | | unemployment rate | - | Measuring the employment status of the population. |
| | | pension insurance participation rate | + | Measurement of the population's insurance status. |
| | | health insurance participation rate | + | |
| | population | total population at year end | + | Measuring total population. |
| | | population density | - | Measuring population congestion. |
| | healthcare | investment in health | + | Measuring the scale of health development. |
| | | number of beds in health-care institutions per 10,000 persons | + | Measuring healthcare resources. |
| | | health technicians per 10,000 population | + | |
| | education | investment in education | + | Measuring the scale of educational development. |
| | | number of students enrolled in general higher education | + | Measuring the level of education. |
| | | teacher-student ratio | + | Measuring educational resources. |
| Ecological Subsystem | environmental carrying capacity | cropland area | + | Measuring arable land resources. |
| | | proportion of days with satisfactory air quality | + | Measuring air quality. |
| | | fertilizer use per unit of arable land area | - | Measuring the quality of arable land. |
| | energy consumption | water use per 10,000 yuan of total production | - | Measuring water consumption. |
| | | energy consumption per 10,000 yuan of industrial output | - | Measuring energy consumption. |
| | | electricity consumption per 10,000 yuan of industrial output | - | Measuring electricity consumption. |

Note

"+" represents a positive indicator, "-" represents a negative indicator.

**2.2.3. Data sources and processing.** The data presented in this paper are sourced from the Statistical Yearbook of Province X for 2011–2021, the Sixth and Seventh National Population Census Bulletin of Province X, and the 2010–2020 Statistical Bulletin of National Economic Development for State N. Missing data exhibiting significant monotonicity were supplemented using linear interpolation, while irregular data were supplemented by averaging. Finally, the Stata software was utilized to calculate weightings for the indicators, and Table 2 displays the results. In addition, the calculation of the ecological niche overlap involved using

**Table 2. Allocation of weights in the ecological niche evaluation system for economic development.**

| System Layer | Normative Layer | Indicator Layer | Indicator Weights |
|---|---|---|---|
| Economic Subsystem | economic aggregate | total regional production | 0.0166 |
| | | fixed-asset investment | 0.0103 |
| | | unit yield of food crops | 0.0488 |
| | | industrial output per capita | 0.0678 |
| | | tourism revenue | 0.1418 |
| | economic structure | share of tertiary output in total production | 0.0098 |
| | | total retail sales of consumer goods as a share of total production | 0.0390 |
| | | percentage of population in agriculture | 0.0317 |
| Social Subsystem | living standards of the population | disposable income of the population | 0.0333 |
| | | unemployment rate | 0.0115 |
| | | pension insurance participation rate | 0.0200 |
| | | health insurance participation rate | 0.0664 |
| | population | total population at year end | 0.0367 |
| | | population density | 0.0221 |
| | healthcare | investment in health | 0.0511 |
| | | number of beds in health-care institutions per 10,000 persons | 0.0698 |
| | | health technicians per 10,000 population | 0.0258 |
| | education | investment in education | 0.0502 |
| | | number of students enrolled in general higher education | 0.0565 |
| | | teacher-student ratio | 0.0305 |
| Ecological Subsystem | environmental carrying capacity | cropland area | 0.0639 |
| | | proportion of days with satisfactory air quality | 0.0254 |
| | | fertilizer use per unit of arable land area | 0.0182 |
| | energy consumption | water use per 10,000 yuan of total production | 0.0344 |
| | | energy consumption per 10,000 yuan of industrial output | 0.0138 |
| | | electricity consumption per 10,000 yuan of industrial output | 0.0047 |

ArcGIS software to identify the spatial centre coordinates of the four regions, straight-line distances were then determined between these regions. As area Z is the nearest to area S, this distance was used as the unit of measurement to determine the relative distances between the other areas.

## 3. Results and discussion

### 3.1. Evaluation of ecological niche width for economic development

According to the constructed ecological niche evaluation system, the ecological niche width of each subsystem and the comprehensive ecological niche width in the four regions from 2011 to 2020 were calculated by Eqs (1) and (2), and the results are shown in Table 3.The findings indicate a fluctuating upward trend in the ecological niche widths across all four regions, with the most significant growth in region T, which increased by 120.3% between 2011 and 2020. Table 4 illustrates the division of regional grades into four levels based on the values of ecological niche width. The ecological niche width composite score for state N, from 2011 to 2020, was graded using the criteria specified. Regions K and S have comparable ecological niche widths and both will be designated as class II areas in 2020, an improvement from their previous class III status. Conversely, regions Z and T will be designated as class III areas in 2020, having previously been class IV areas between 2013 and 2016. Region Z was a class III area

**Table 3. Ecological niche widths of economic development subsystems in the four regions of the state N from 2011 to 2020.**

| Region | Year | Economic Subsystem | Social Subsystem | Ecological Subsystem | Synthesis |
|---|---|---|---|---|---|
| Region K | 2020 | 0.7907 | 0.4242 | 0.6570 | 0.5956 |
| | 2019 | 0.3173 | 0.3724 | 0.5256 | 0.3768 |
| | 2018 | 0.2912 | 0.4097 | 0.6164 | 0.3995 |
| | 2017 | 0.2901 | 0.4088 | 0.6409 | 0.4026 |
| | 2016 | 0.2272 | 0.2002 | 0.3627 | 0.2361 |
| | 2015 | 0.2536 | 0.2080 | 0.4849 | 0.2691 |
| | 2014 | 0.2115 | 0.2275 | 0.4584 | 0.2587 |
| | 2013 | 0.2397 | 0.2182 | 0.3810 | 0.2522 |
| | 2012 | 0.2413 | 0.2023 | 0.5109 | 0.2661 |
| | 2011 | 0.4634 | 0.2105 | 0.5953 | 0.3647 |
| Region Z | 2020 | 0.2809 | 0.5704 | 0.3902 | 0.4356 |
| | 2019 | 0.2520 | 0.4035 | 0.4903 | 0.3620 |
| | 2018 | 0.1607 | 0.2787 | 0.4432 | 0.2619 |
| | 2017 | 0.2117 | 0.2947 | 0.5223 | 0.3009 |
| | 2016 | 0.1622 | 0.2409 | 0.3332 | 0.2269 |
| | 2015 | 0.1719 | 0.1973 | 0.4245 | 0.2244 |
| | 2014 | 0.1693 | 0.2421 | 0.2346 | 0.2142 |
| | 2013 | 0.2006 | 0.1895 | 0.2551 | 0.2041 |
| | 2012 | 0.2389 | 0.2209 | 0.4387 | 0.2624 |
| | 2011 | 0.2411 | 0.2339 | 0.3323 | 0.2523 |
| Region S | 2020 | 0.7188 | 0.5315 | 0.5506 | 0.6031 |
| | 2019 | 0.2181 | 0.4149 | 0.6012 | 0.3728 |
| | 2018 | 0.3990 | 0.3817 | 0.4837 | 0.4044 |
| | 2017 | 0.3378 | 0.4638 | 0.7944 | 0.4707 |
| | 2016 | 0.2335 | 0.2803 | 0.2914 | 0.2650 |
| | 2015 | 0.2447 | 0.2956 | 0.4134 | 0.2958 |
| | 2014 | 0.2078 | 0.2215 | 0.4774 | 0.2575 |
| | 2013 | 0.2530 | 0.2706 | 0.3109 | 0.2706 |
| | 2012 | 0.3024 | 0.2474 | 0.5112 | 0.3098 |
| | 2011 | 0.3573 | 0.2659 | 0.3499 | 0.3128 |
| Region T | 2020 | 0.4399 | 0.5553 | 0.4415 | 0.4948 |
| | 2019 | 0.1309 | 0.3651 | 0.4982 | 0.3008 |
| | 2018 | 0.1449 | 0.3342 | 0.4300 | 0.2803 |
| | 2017 | 0.1515 | 0.4679 | 0.5654 | 0.3678 |
| | 2016 | 0.0741 | 0.2952 | 0.2540 | 0.2077 |
| | 2015 | 0.0979 | 0.2365 | 0.2946 | 0.1951 |
| | 2014 | 0.0936 | 0.2126 | 0.2619 | 0.1770 |
| | 2013 | 0.1240 | 0.2577 | 0.2001 | 0.1996 |
| | 2012 | 0.1533 | 0.2401 | 0.3931 | 0.2329 |
| | 2011 | 0.1885 | 0.2359 | 0.2738 | 0.2246 |

while region T was class IV in 2011 and 2012, which distinguishes them. The ecological niche widths of all regions, however, demonstrate significant improvement in 2020.

The different dimensions of development in state N are graded through the evaluation of subsystems. The first aspect evaluated is the ecological niche width of the economic subsystems. Both the economic subsystems of region K and S have been unstable, fluctuating between class III and IV during 2011 and 2019. However, in 2020, the economic subsystem of

**Table 4. Ecological niche width classifications.**

| Class | Ecological Niche Width | Instruction |
|---|---|---|
| I | $0.75 < M_i \leq 1$ | The greater the value, the higher the economic development status of the region and the stronger its regional influence. |
| II | $0.5 < M_i \leq 0.75$ | |
| III | $0.25 < M_i \leq 0.5$ | |
| IV | $0 \leq M_i \leq 0.25$ | |

region K reached class I, while the region S remained at class II. Compared to the first two regions, the economic subsystems of regions Z and T are relatively more stable, yet also comparatively underdeveloped, both classified as class IV until 2018. Region Z reached class III in 2019, while region T lagged behind by a year, reaching class III in 2020. Additionally, the ecological niche width of the social subsystem is assessed. Compared with the other three regions, the social subsystem of region S exhibits greater instability, oscillating between classes III and IV from 2011 to 2019. The social subsystem of region K is marginally inferior in comparison to the other three prefectures. In 2020, the ecological niche width of the social subsystems of the other three regions attains class II, while the region K remains at class III. Thirdly, this study evaluates the ecological niche width of different ecological subsystems. The ecological subsystems of state N display no consistent historical trends. In 2020, region K and S reached class II, while both region Z and T were at class III.

In short, between identical subsystems in different areas, as well as between different subsystems within the same region, there are differences in evolution of time trends. Overall, the ecological niche width of the four regions in state N is at a low level, with the combined economic development status of regions K and S slightly higher than that of regions Z and T.

Based on the average value of ecological niche width over a 10-year period, as presented in Table 5, the results indicate that the ecological niches have a comprehensive width with a structural characteristic of "low level and low gap". The order of economic development from highest to lowest is region S, region K, region Z and region T. These four regions are all classified as class III, with low economic development status and weak regional influence and dominance. Starting from various subsystem perspectives, the ecological niche widths of the regions were ranked. The economic subsystems were ranked in descending order as region K, region S, region Z, and region T. The social subsystems were ranked in descending order as region S, region T, region K, and region Z. The ecological subsystem is ordered from highest to lowest as region K, region S, region Z, and region T. In conclusion, the level of regional economic development's ecological niche width is affected comprehensively by the spatial and temporal correlation among the economic subsystems, social subsystems, and ecological subsystems. Region K's ecological niche width for economic and ecological subsystems is slightly higher than that of region S. However, the lower ecological niche width of social subsystems in region K results in the ecological niche composite width being surpassed by region S.

**Table 5. Mean ecological niche widths of economic development subsystems in the four regions of the state N.**

| Region | Region K | Region Z | Region S | Region T |
|---|---|---|---|---|
| Economic Subsystem | 0.3326 | 0.2089 | 0.3272 | 0.1599 |
| Social Subsystem | 0.2882 | 0.2872 | 0.3373 | 0.3201 |
| Ecological Subsystem | 0.5233 | 0.3864 | 0.4784 | 0.3613 |
| Synthesis | 0.3421 | 0.2745 | 0.3563 | 0.2681 |

**Table 6. Ecological niche overlap for economic development between the two regions of state N from 2011 to 2020.**

| Year | Region K and Z | Region K and S | Region K and T | Region Z and S | Region Z and T | Region S and T |
|------|----------------|----------------|----------------|----------------|----------------|----------------|
| 2020 | 0.2791 | 0.3108 | 0.4618 | 0.6857 | 0.2922 | 0.4809 |
| 2019 | 0.3335 | 0.3379 | 0.2209 | 0.9336 | 0.2017 | 0.3983 |
| 2018 | 0.3304 | 0.2955 | 0.3746 | 0.8015 | 0.2750 | 0.5009 |
| 2017 | 0.3345 | 0.3154 | 0.4100 | 0.8940 | 0.3096 | 0.4989 |
| 2016 | 0.3292 | 0.3332 | 0.4187 | 0.9284 | 0.3168 | 0.4865 |
| 2015 | 0.3321 | 0.3601 | 0.3877 | 0.9286 | 0.2957 | 0.4144 |
| 2014 | 0.3299 | 0.3516 | 0.4434 | 0.9338 | 0.3224 | 0.4907 |
| 2013 | 0.3228 | 0.3481 | 0.4438 | 0.9816 | 0.3385 | 0.4954 |
| 2012 | 0.3157 | 0.3163 | 0.4353 | 0.9706 | 0.3437 | 0.4974 |
| 2011 | 0.3298 | 0.3760 | 0.4668 | 0.9717 | 0.3282 | 0.4760 |

## 3.2. Evaluation of ecological niche overlap for economic development

According to the ecological niche assessment system, Eqs (3)–(5) were used to calculate the ecological niche overlap for economic development between each of the regions in State N from 2011 to 2020. The resulting data is presented in Table 6. Based on the value of the ecological niche overlap degree, the regional grade is classified into four levels, illustrated in Table 7. The findings indicate that economic development overlap between region Z and region S was the highest, measuring above 0.75 from 2011 to 2019, achieving a class I competition ranking. The level of competition decreased in 2020, yet remained at the class II ranking. The primary cause of this phenomenon is the substantial overlap in resource usage between Regions Z and S due to their geographic closeness. The competing level between other areas is generally low, usually ranking at class III or IV. In addition, with the exception of the slight increase in ecological niche overlap between region S and T over the 10-year period, the ecological niche overlap of all other regions has shown a fluctuating downward trend. This shows that the existence of the phenomenon of competition has been noted in the process of economic development, and that different regions have developed in different directions in order to avoid competition.

Based on Table 8, the different dimensions of development in state N are graded through the evaluation of subsystems. The first aspect evaluated is the ecological niche overlap of the economic subsystems. Between 2011 and 2020, the economic subsystem in region S fluctuated between class II and III. In contrast, the economic subsystems in the other three regions remained stable, all remaining at the same level over the ten-year period, with the difference that regions K and T remained at class III, while the region Z remained at class II. Additionally, the ecological niche overlap of the social subsystem is assessed. Between 2011 and 2020, the social subsystems in the region S were more unstable and had a higher degree of overlap compared to the other three regions, oscillating between class II and III. In contrast, the overlap of social subsystems in the other three regions remained at class III. Thirdly, this study evaluates

**Table 7. Ecological niche overlap classifications.**

| Class | Ecological Niche Overlap | Instruction |
|-------|--------------------------|-------------|
| I | $0.75 < M_i \leq 1$ | The greater the value, the stronger the competition between regions. |
| II | $0.5 < M_i \leq 0.75$ | |
| III | $0.25 < M_i \leq 0.5$ | |
| IV | $0 \leq M_i \leq 0.25$ | |

**Table 8. Ecological niche overlaps of economic development subsystems in the four regions of the state N from 2011 to 2020.**

| Region | Year | Economic Subsystem | Social Subsystem | Ecological Subsystem | Synthesis |
|---|---|---|---|---|---|
| Region K | 2020 | 0.3717 | 0.3599 | 0.3143 | 0.3506 |
| | 2019 | 0.2966 | 0.3547 | 0.3533 | 0.2975 |
| | 2018 | 0.3351 | 0.3494 | 0.3379 | 0.3335 |
| | 2017 | 0.3554 | 0.3619 | 0.3539 | 0.3533 |
| | 2016 | 0.3735 | 0.3423 | 0.3447 | 0.3603 |
| | 2015 | 0.3919 | 0.3468 | 0.2769 | 0.3600 |
| | 2014 | 0.3832 | 0.3589 | 0.3131 | 0.3750 |
| | 2013 | 0.3756 | 0.3630 | 0.3432 | 0.3716 |
| | 2012 | 0.3576 | 0.3634 | 0.3314 | 0.3558 |
| | 2011 | 0.3984 | 0.3705 | 0.3908 | 0.3908 |
| Region Z | 2020 | 0.4787 | 0.4332 | 0.4353 | 0.4190 |
| | 2019 | 0.4963 | 0.4587 | 0.4221 | 0.4896 |
| | 2018 | 0.4778 | 0.4381 | 0.4005 | 0.4689 |
| | 2017 | 0.5213 | 0.4279 | 0.3311 | 0.5127 |
| | 2016 | 0.5403 | 0.4101 | 0.4588 | 0.5248 |
| | 2015 | 0.5414 | 0.4132 | 0.3913 | 0.5188 |
| | 2014 | 0.5447 | 0.3919 | 0.4457 | 0.5287 |
| | 2013 | 0.5559 | 0.4331 | 0.4649 | 0.5476 |
| | 2012 | 0.5478 | 0.4608 | 0.4670 | 0.5433 |
| | 2011 | 0.5560 | 0.4531 | 0.5432 | 0.5432 |
| Region S | 2020 | 0.5251 | 0.5356 | 0.5056 | 0.4925 |
| | 2019 | 0.5625 | 0.5502 | 0.5055 | 0.5566 |
| | 2018 | 0.5386 | 0.5274 | 0.4686 | 0.5326 |
| | 2017 | 0.5755 | 0.5308 | 0.4562 | 0.5694 |
| | 2016 | 0.5972 | 0.4965 | 0.5549 | 0.5827 |
| | 2015 | 0.6012 | 0.5117 | 0.4294 | 0.5677 |
| | 2014 | 0.6073 | 0.4870 | 0.4890 | 0.5920 |
| | 2013 | 0.6172 | 0.5139 | 0.5020 | 0.6084 |
| | 2012 | 0.5996 | 0.5167 | 0.5202 | 0.5948 |
| | 2011 | 0.6189 | 0.5168 | 0.6079 | 0.6079 |
| Region T | 2020 | 0.4273 | 0.3918 | 0.3597 | 0.4116 |
| | 2019 | 0.2622 | 0.4114 | 0.4079 | 0.2737 |
| | 2018 | 0.3842 | 0.4022 | 0.3343 | 0.3835 |
| | 2017 | 0.4130 | 0.3944 | 0.3770 | 0.4061 |
| | 2016 | 0.4267 | 0.3641 | 0.3801 | 0.4074 |
| | 2015 | 0.4330 | 0.3926 | 0.2657 | 0.3659 |
| | 2014 | 0.4282 | 0.3979 | 0.3708 | 0.4188 |
| | 2013 | 0.4338 | 0.4052 | 0.3977 | 0.4259 |
| | 2012 | 0.4299 | 0.4069 | 0.3807 | 0.4255 |
| | 2011 | 0.4302 | 0.3938 | 0.4236 | 0.4236 |

the ecological niche overlap of different ecological subsystems. The ecological niche subsystem hierarchy is comparable to that of social subsystems. Region S is more unstable and has a higher degree of overlap, oscillating between class II and III from 2011 to 2020. The overlap of social subsystems in the other three regions is primarily maintained at class III, with only region Z at class II in 2011.

**Table 9. Mean ecological niche overlaps of economic development subsystems in the four regions of the state N.**

| Region | Region K | Region Z | Region S | Region T |
|---|---|---|---|---|
| Economic Subsystem | 0.3639 | 0.5260 | 0.5843 | 0.4069 |
| Social Subsystem | 0.3571 | 0.4320 | 0.5187 | 0.3960 |
| Ecological Subsystem | 0.3360 | 0.4360 | 0.5039 | 0.3698 |
| Synthesis | 0.3548 | 0.5097 | 0.5705 | 0.3942 |

Table 8 shows the combined overlap of the different regions. Notably, regions K and T display low overlap while the ecological niche's comprehensive overlap during the 10-year period is at class III. Region Z displays high overlap from 2011 to 2017, with the ecological niche's comprehensive overlap at class II, which declines to class III from 2018 to 2020. In contrast, region S demonstrates high overlap from 2011 to 2019, with the ecological niche's comprehensive overlap at class II, which only reduces to class III in 2020.

Based on the average value of ecological niche overlap over a 10-year period, as presented in Table 9, the results indicate that the ecological niche overlap is significantly influenced by spatial factors, region Z and S have a high ecological niche overlap due to their close spatial proximity. The order of the competition degree in economic development, from low to high, is region K, region T, region Z, and region S. The ecological niche overlap is low in region K and T, at level III, as well as high in region Z and S, at level II, but the latter two are gradually moving towards lower overlap. Starting from various subsystem perspectives, the ecological niche widths of the regions were ranked. The rankings of ecological niche overlap for the subsystems were consistent with the combined ecological niche overlap ranking, which is region K, region T, region Z, and region S. However, the geographic proximity of regions Z and S results in excessive competition between them, which influences the combined ecological niche overlap of these regions.

## 4. Conclusions and recommendations

### 4.1. Conclusions

Based on previous research into evaluating economic development, this paper presents a case study of the borderland state N. The study applies the ecological niche theory to measure the width and overlap of the economic development ecological niche of the four regions within it. The study then evaluates the economic development status of state N, as well as the level of competition. The main conclusions are as follows:

Firstly, tourism is the primary factor influencing the ecological niche of economic development of state N. Table 2 shows the weight allocation of each indicator in the economic development ecological niche index system, as determined by the entropy value method used in this paper. Only one indicator, tourism income, has a weight of more than 10%, reaching 14.18%, while the weights of the other indicators are all below 10%. Secondly, the economic development's ecological niche width in state N exhibits the structural characteristics of "low level and low gap". According to the ecological niche width model, the average ecological niche width value of economic development in each region is at class III, indicating a lower development status and weaker regional influence and dominance. The level of regional economic development's ecological niche width is affected comprehensively by the spatial and temporal correlation among the economic subsystems, social subsystems, and ecological subsystems. Thirdly, the spatial factors greatly influence the degree of overlap of the economic development ecological niche of state N. The ecological niche overlap model was used to measure the degree of overlap between economic development in each region. The geographic proximity of regions

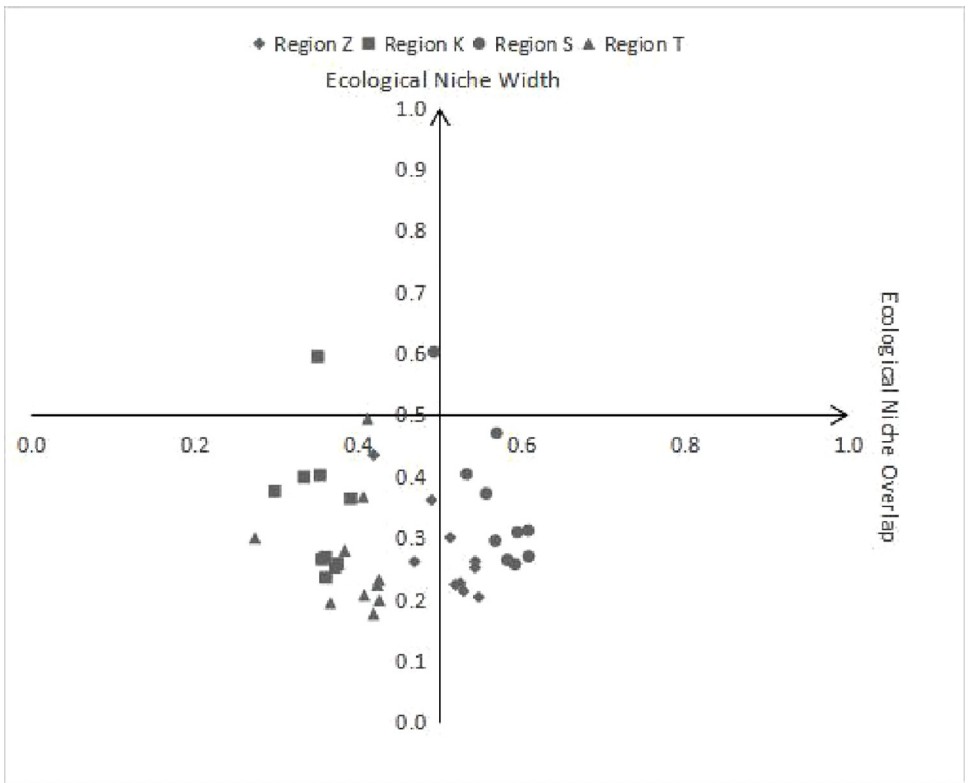

**Fig 2. Scatterplot of width and overlap of ecological niches for economic development in the four regions of the state N.**

Z and S results in excessive competition between them, which influences the combined ecological niche overlap of these regions.

To accurately portray the complete status of ecological niche width and overlap in each region of State N, a coordinate system is created with 0.5 as the dividing line. The ecological niche overlap serves as the horizontal axis, while the ecological niche width serves as the vertical axis. The figures of the ecological niche width and overlap of the four regions from 2011 to 2020 are marked on the graph to produce Fig 2. In terms of the combined ecological niche width and overlap, regions K and T exhibit a comparable economic development status, with most data points situated in quadrant 3, highlighting "low width and low overlap". Similarly, regions S and Z share a comparable economic development status, with most data points located in quadrant 4, denoting "low width and high overlap".

## 4.2. State N economic development strategy

The economic development of State N is classified, based on the ecological niche theory relationship shown in Fig 1. Different development strategies must be employed for distinct regions. Regions S and Z belong to the "low-width, high-overlap" region and call for a specialized separation strategy. On the other hand, regions K and T belong to the "low-width, low-overlap" field and require strengthened expansion tactics.

**4.2.1. Specialized separation.** Due to the "low-width and high-overlap" characteristics, regions S and Z should implement ecological niche specialized separation in economic development. This requires these regions to leverage their local economic development advantages, strive to explore the region's development potential, and achieve ecological niche separation

from other regions through precise positioning. This will enable the implementation of a gap-type development strategy and create more space for the growth and survival of all stakeholders.

Region S is in the early stages of industrialization, with a weak industrial base. However, the region has the necessary conditions and resources for the development of new industries, such as oil and gas, green mining, new energy, and new materials. Therefore, Region S should leverage its location advantages and industrial base, expedite local oil and gas well exploration, support industrial park construction, promote comprehensive natural gas utilization, introduce and establish natural gas chemical industries, enhance new energy advantages, and accelerate the establishment of a modern industrial system. Furthermore, region S possesses abundant solar energy resources. Therefore, it is important to develop the photovoltaic industry and cultivate new forms of clean energy. This will promote the energy revolution and accelerate the planning and construction of a new energy system, ultimately reducing fossil fuel consumption and pollution. Region S has a high-quality new energy resource advantage, which serves as the foundation for the development of new energy-related industries. This strategy aims to achieve economic growth, reduce energy consumption, and decrease pollution. It is a specialized separation approach for the region S's development.

Region Z is geographically closer to region S and has a more mature agro-industrial system, but it is not as rich in new energy resources as region S. To reduce the overlap in resource use between the two regions, region Z can optimize its agro-industry, create a brand image for its speciality industries, and promote a high degree of agglomeration of factors of production in these industries to form an industrial and brand advantage. In addition, the integration of agriculture and tourism has extended the development of the agricultural industry beyond the primary sector to a more integrated approach involving both primary and tertiary industries. Therefore, region Z should integrate the agricultural speciality industry and tourism industry in an organic manner, and develop speciality agro-tourism products that are tailored to local conditions, in order to achieve a mutually beneficial outcome for both industries. Region Z has a development advantage in its mature agricultural industry system. The integration of agriculture and tourism is a key pathway for development, which can improve the efficiency of the agricultural industry and expand its development opportunities. This is achieved through the specialized separation strategy of region Z.

**4.2.2. Strengthened expansion.** Regions K and T are characterized as having a "low-width, high-overlap" feature, and as such, they should implement the ecological niche strengthened expansion strategy in their economic development, with the main objective being the filling of shortfalls. This includes expanding potential ecological niches or introducing new ones.

In region K, the basic public service system needs improvement to strengthen the social subsystem. It is important to focus on promoting social security in the region. Improving healthcare and accelerating the construction of a high-quality education system are necessary steps to improve the ecological status of the social subsystem. A more proactive employment policy should also be implemented. In addition, promoting self-employment, attracting capital and talent from outside the region, and fostering entrepreneurship-led employment are important measures to enhance the ecological status of the social subsystem in region K by introducing new ecological niches. According to the above, the development of region K falls short in the area of social security. To improve the ecological status of region K, an strengthened expansion strategy is proposed, which includes expanding potential ecological niches and introducing new ones. The former involves improving medical care and education levels, while the latter aims to attract more population through policies.

Improving the economic development environment and optimizing the economic structure is an important step for region T to enhance its ecological position. Region T has a thriving agricultural economy, but its industrial sector lags behind, with a dominance of primary sector industries. Region T has a plentiful population, so the development of industries should focus on labour-intensive industries. Additionally, due to the region's excellent agricultural resources, there is potential for the vigorous development of the deep processing industry of agricultural products. This can create a whole industrial chain pattern of agricultural product processing that integrates high-quality raw material bases, characteristics of industrial clusters, and productive service platforms. The result will be a green and organic agricultural product processing base. In addition, labour-intensive industries such as electronics assembly, textile and garment production, and ethnic handicrafts can also contribute to the development of the industrial economy in the region T. Based on the above, the economic shortcomings of the development of region T are primarily in the industrial sector. Given its abundant agricultural resources and labour force, prioritizing the development of labour-intensive industries could be an strengthened expansion strategy to enhance the region's economy.

## Supporting information

**S1 Dataset.**
(XLSX)

## Acknowledgments

Throughout the writing of this dissertation I have received a great deal of support and assistance. I would first like to thank my supervisor, Nurguli Abdusuli, whose expertise was invaluable in formulating the research questions and methodology. Your insightful feedback pushed me to sharpen my thinking and brought my work to a higher level. In addition, I would like to thank my parents for their wise counsel and sympathetic ear. You are always there for me.

## Author Contributions

**Conceptualization:** Nurguli Abdusuli.

**Data curation:** Hang Zhang.

**Formal analysis:** Hang Zhang.

**Software:** Hang Zhang.

**Supervision:** Nurguli Abdusuli.

**Validation:** Nurguli Abdusuli.

**Writing – original draft:** Hang Zhang.

**Writing – review & editing:** Nurguli Abdusuli.

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
