## [Decision Letter · Decision Letter 0]

26 Feb 2024

PONE-D-24-01398Ecological Niche Measurement and High-quality Development of "the Belt and Road" Core AreaPLOS ONE

Dear Dr. Zhang,

Thank you for submitting your manuscript to PLOS ONE. After careful consideration, we feel that it has merit but does not fully meet PLOS ONE’s publication criteria as it currently stands. Therefore, we invite you to submit a revised version of the manuscript that addresses the points raised during the review process.

We look forward to receiving your revised manuscript.

Kind regards,

Tunira Bhadauria, Ph.D.

Academic Editor

PLOS ONE

“The National Social Science Fund of China: Study on the Cultural Coexistence Paradigm of Embeddedness Multi-ethnicof Villages in the Context of "Culture Moistening Xinjiang", grant number 22BMZ123.”

4. We notice that your supplementary figures are uploaded with the file type 'Figure'. Please amend the file type to 'Supporting Information'. Please ensure that each Supporting Information file has a legend listed in the manuscript after the references list.

Reviewers' comments:

Reviewer's Responses to Questions

**Comments to the Author**

1. Is the manuscript technically sound, and do the data support the conclusions?

Reviewer #1: Yes

Reviewer #2: Yes

2. Has the statistical analysis been performed appropriately and rigorously? 

Reviewer #1: Yes

Reviewer #2: Yes

3. Have the authors made all data underlying the findings in their manuscript fully available?

Reviewer #1: No

Reviewer #2: Yes

4. Is the manuscript presented in an intelligible fashion and written in standard English?

Reviewer #1: No

Reviewer #2: Yes

5. Review Comments to the Author

Reviewer #1: Greetings! This manuscript introduces the ecological niche theory to regional development assessment and conducts a study using Xinjiang (South Xinjiang), one of the core regions of the Belt and Road, as the study area. The results are significant for the coordinated development of the South Xinjiang, as well as for the better implementation of the Belt and Road Initiative. However, there are some problems that required systematic revision before the manuscript accepted.

(1) Title appropriate? In terms of content, the article takes the South Xinjiang as the study area, while the title is the Belt and Road Core Area. Is the case area typical enough to represent the Belt and Road Core Area?

(2) Abstract: inconsistent font size.

(3) Introduction: The first paragraph lacks support from relevant literature.

(4) Introduction: In the first paragraph, revised "One Belt, One Road" initiative to the Belt and Road Initiative.

(5) Methods: Are the 26 indicators selected in Table1 tested for covariance?

(6) Conclusion: Too long, streamlining is required.

(7) Recommendation section (4.2): Since this article does not analyze the causes of interregional development disparities, the section is general and unfocused. It is recommended that this section be revised.

Reviewer #2: The author applies niche theory to the evaluation of high-quality development of regional economy. From the theoretical level, the application, nesting and transfer of niche concept are relatively smooth, and the theoretical explanation is clear. From the perspective of article writing, the author 's writing logic is more rigorous, the expression is clear, and the words are rigorous and easy to understand. In general, the quality of the article is high, but there are still some problems, and it is recommended to modify it slightly. The specific questions are as follows :

1.The research results of the abstract need to be expressed in detail, and the author 's description is too brief. For example, in what range does the width value represent which areas in southern Xinjiang show this structural feature ? There are differences in the degree of overlap, what kind of difference, whether it can be quantified ? Suggested modification

2.In the third paragraph of the introduction, the concept of niche is introduced here, but the application of niche in high-quality economic development has made little progress, and the direct transition to the evaluation of economic development with niche theory requires an entry point.

3.In the second part, the first part of the materials and methods : Niche theory. This part can also be placed in the research progress, and it is not suitable for materials and methods. It does not belong to the material, and the niche theory does not belong to the method. Here is also the definition and connotation of the niche.

4.In the discussion and conclusion part, we should add some discussion methods and the accuracy of the results.

5.References should be properly updated, plus some recent literature.

6. PLOS authors have the option to publish the peer review history of their article (what does this mean?). If published, this will include your full peer review and any attached files.

Reviewer #1: No

Reviewer #2: No

---

## [Author Response · Author response to Decision Letter 0]

18 Mar 2024

We gratefully thank the editor and all reviewers for their time spend making their constructive remarks and useful suggestions, which has significantly raised the quality of the manuscript and has enable us to improve the manuscript. Each suggested revision and comment, brought forward by the reviewers was accurately considered. Below the comments of the reviewers are response point by point and the revisions are indicated.

Reviewer 1

Comment 1：Title appropriate? In terms of content, the article takes the South Xinjiang as the study area, while the title is the Belt and Road Core Area. Is the case area typical enough to represent the Belt and Road Core Area?

The title was decided by consensus among the authors and is in line with current research. The establishment of the Belt and Road Core Area fulfills three necessary conditions: obvious geographical advantages, superior resource endowment and insufficient economic development. First of all, South Xinjiang is located in China's western border, in the "Belt and Road" along the strategic corridor and open gateway for the interaction of countries, with Khunjerab, Turgat, Пункт пропус and other national first-class ports, geographical advantages and location advantages are very obvious, and can be done with other countries to achieve direct interconnection and interaction is the "Belt and Road" all resources, factors, exchange and interaction must pass through the gateway. "Belt and Road" all resources, factors, exchange and interaction must pass through the gateway. Secondly, the Southern Xinjiang has extensive ties with Central Asia and unique advantages in terms of language, religion, and economic influence. It has the necessary conditions for building the core area in terms of infrastructure, economic and trade cooperation, and humanistic exchanges. Finally, the economy of the Southern Xinjiang exhibits imbalances and inadequacies. There is a clear need for improvement in economic development, Southern Xinjiang could establish a more common language and cooperation basis with countries along the “Belt and Road”. Based on the above, the Southern Xinjiang shares both the advantages and disadvantages of the core regions of the Belt and Road. This makes it a typical representative of the core regions of the Belt and Road.

Comment 2: Abstract: inconsistent font size.

Changed in the text.

Comment 3: Introduction: The first paragraph lacks support from relevant literature.

The introduction's first paragraph provides a concise overview of the Southern Xinjiang's current status. Research on the Belt and Road Core Area is primarily conducted at a macro or provincial level, with limited focus on specific regions. There is limited literature available on the current state of development of Southern Xinjiang. However, the first paragraph of the introduction is not unfounded. It accurately reflects the strengths, weaknesses, and current state of development of the region, as taken from policy documents of local governments in China. The references for these documents are included in the cited literature section.

Comment 4: Introduction: In the first paragraph, revised "One Belt, One Road" initiative to the Belt and Road Initiative.

Changed in the text.

Comment 5: Methods: Are the 26 indicators selected in Table1 tested for covariance?

Based on the significance of covariance and current research on the evaluation of economic ecological niches, we believe that testing for covariance of the indicators in Table 1 is unnecessary. In this paper, we calculated the economic ecological niche as a quantitative result by synthesizing 26 variables. As the dependent variable is unobservable, the covariance test is not applicable for this study. Furthermore, in the existing literature on economic ecological niches, the indicator evaluation system is often not tested for covariance.

Comment 6: Conclusion: Too long, streamlining is required.

The data analyses and discussions have been moved from the conclusions to section Results and discussion (3) to improve clarity and conciseness.

Comment 7: Recommendation section (4.2): Since this article does not analyze the causes of interregional development disparities, the section is general and unfocused. It is recommended that this section be revised.

In sections 3.1 and 3.2, we presented the results of each economic development subsystem to support our recommendations. We also revised the recommendations according to the strengths and weaknesses of the local economic development, ensuring alignment with the development goals of each region.

Reviewer 2

Comment 1: The research results of the abstract need to be expressed in detail, and the author 's description is too brief. For example, in what range does the width value represent which areas in southern Xinjiang show this structural feature ? There are differences in the degree of overlap, what kind of difference, whether it can be quantified ? Suggested modification.

The conclusions in the summary section were expanded based on modifications made to the results in the conclusions section. This was achieved by adding the hierarchical level of ecological width or overlap and characteristics to provide a more comprehensive analysis.

Comment 2: In the third paragraph of the introduction, the concept of niche is introduced here, but the application of niche in high-quality economic development has made little progress, and the direct transition to the evaluation of economic development with niche theory requires an entry point.

The last paragraph of the introduction now begins with a section that introduces the advantages and uniqueness of using ecological niche theory to study economic development. This addition improves the logical structure of the text.

Comment 3: In the second part, the first part of the materials and methods : Niche theory. This part can also be placed in the research progress, and it is not suitable for materials and methods. It does not belong to the material, and the niche theory does not belong to the method. Here is also the definition and connotation of the niche.

The ecological niche theory section of materials and methods was divided into two parts. The first part covers the theoretical description and is placed after the ecological niche definition and connotation in the third paragraph of the introduction. The second part includes only the ecological niche width model and the ecological niche overlap model, which are still part of the research methodology and are retained in section 2.1.

Comment 4: In the discussion and conclusion part, we should add some discussion methods and the accuracy of the results.

The comments from multiple reviewers were consolidated, and as a result, the data analyses and discussion were relocated from the conclusion to section 3 to enhance the conclusion's clarity and conciseness.

Comment 5: References should be properly updated, plus some recent literature.

The article cites literature from the last five years (2020-2024), which accounts for approximately 75% of the total literature. To increase this percentage to over 80%, a few additional literature were included.

---

## [Editor Report · Decision Letter 1]

8 Apr 2024

Ecological Niche Measurement and High-quality Development of "the Belt and Road" Core Area

PONE-D-24-01398R1

Dear Dr. Zhang

We’re pleased to inform you that your manuscript has been judged scientifically suitable for publication and will be formally accepted for publication once it meets all outstanding technical requirements.

Kind regards,

Tunira Bhadauria, Ph.D.

Academic Editor

PLOS ONE
---

## [Editor Report · Acceptance letter]

30 Apr 2024

PONE-D-24-01398R1 

PLOS ONE

Dear Dr. Zhang, 

I'm pleased to inform you that your manuscript has been deemed suitable for publication in PLOS ONE. Congratulations! Your manuscript is now being handed over to our production team.

Kind regards, 

on behalf of

Dr. Tunira Bhadauria 

Academic Editor

PLOS ONE